# Phosphatidic Acid Stimulates Lung Cancer Cell Migration through Interaction with the LPA1 Receptor and Subsequent Activation of MAP Kinases and STAT3

**DOI:** 10.3390/biomedicines11071804

**Published:** 2023-06-23

**Authors:** Ana Gomez-Larrauri, Patricia Gangoiti, Laura Camacho, Natalia Presa, Cesar Martin, Antonio Gomez-Muñoz

**Affiliations:** 1Department of Biochemistry and Molecular Biology, Faculty of Science and Technology, University of the Basque Country (UPV/EHU), P.O. Box 644, 48980 Bilbao, Bizkaia, Spain; anagola@hotmail.com (A.G.-L.); pgangoiti@hotmail.com (P.G.); laura_mond@hotmail.com (L.C.); npretor@gmail.com (N.P.); cesar.martin@ehu.eus (C.M.); 2Respiratory Department, Cruces University Hospital, 48903 Barakaldo, Bizkaia, Spain; 3Department of Molecular Biophysics, Biofisika Institute, University of Basque Country and Consejo Superior de Investigaciones Científicas (UPV/EHU, CSIC), 48940 Leioa, Bizkaia, Spain

**Keywords:** phosphatidic acid, lysophosphatidic acid receptors, lung cancer cell migration, mitogen-activated protein kinases, Janus kinase, signal transducer and activator of transcription

## Abstract

Phosphatidic acid (PA) is a key bioactive glycerophospholipid that is implicated in the regulation of vital cell functions such as cell growth, differentiation, and migration, and is involved in a variety of pathologic processes. However, the molecular mechanisms by which PA exerts its pathophysiological actions are incompletely understood. In the present work, we demonstrate that PA stimulates the migration of the human non-small cell lung cancer (NSCLC) A549 adenocarcinoma cells, as determined by the transwell migration assay. PA induced the rapid phosphorylation of mitogen-activated protein kinases (MAPKs) ERK1-2, p38, and JNK, and the pretreatment of cells with selective inhibitors of these kinases blocked the PA-stimulated migration of cancer cells. In addition, the chemotactic effect of PA was inhibited by preincubating the cells with pertussis toxin (PTX), a Gi protein inhibitor, suggesting the implication of a Gi protein-coupled receptor in this action. Noteworthy, a blockade of LPA receptor 1 (LPA1) with the specific LPA1 antagonist AM966, or with the selective LPA1 inhibitors Ki1645 or VPC32193, abolished PA-stimulated cell migration. Moreover, PA stimulated the phosphorylation of the transcription factor STAT3 downstream of JAK2, and inhibitors of either JAK2 or STAT3 blocked PA-stimulated cell migration. It can be concluded that PA stimulates lung adenocarcinoma cell migration through an interaction with the LPA1 receptor and subsequent activation of the MAPKs ERK1-2, p38, and JNK, and that the JAK2/STAT3 pathway is also important in this process. These findings suggest that targeting PA formation and/or the LPA1 receptor may provide new strategies to reduce malignancy in lung cancer.

## 1. Introduction

Lung cancer is the leading cause of cancer deaths in Western countries, and cases continue to increase despite the fact that the etiology of the disease is well known in the majority of cases [1]. Lung cancer diagnoses are classified into two main groups: small cell lung cancer (SCLC) and non-small cell lung cancer (NSCLC). The latter is the most common type of lung cancer, accounting for over 80% of all lung cancer cases, with adenocarcinoma being the major NSCLC subtype. The progression of lung cancer is associated with a complex network of events involving cell proliferation and migration that ultimately result in the invasion of surrounding or distal tissues to promote secondary tumor formation in the context of metastasis. Most migrating cells have an internal compass that enables them to sense and move along gradients of attracting molecules in the process of chemotaxis [2]. Many chemoattractants, including growth factors and cytokines, stimulate cell migration through an interaction with G-protein-coupled receptors (GPCRs) or receptor tyrosine kinases. These agents, which are mainly of a peptide or protein nature, often cause the activation of lipid-biosynthesizing enzymes that generate intracellular lipid second messengers that control cell proliferation or migration. In particular, the activation of lysophosphatidic acid (LPA) acyltransferase (LPAAT), diacylglycerol (DAG) kinase (DAGK), or phospholipase D (PLD) generates intracellular phosphatidic acid (PA) and autotaxin, a secreted lysophospholipase D enzyme [3], which leads to the formation of LPA from the degradation of lysophosphatidylcholine. The proliferative and chemotactic properties of LPA are well known and have been widely studied in many non-neoplastic and neoplastic cell types [4,5,6,7,8,9,10]. Whilst LPAAT-derived PA in the ER functions as an intermediate in de novo phospholipid and triacylglycerol (TAG) biosynthesis [11], DAGK- or PLD-derived PA produced on other organelle membranes can affect diverse and distinct processes, including actin polymerization, or mTOR signaling [12], which are processes associated with cell migration or proliferation. Although PA has been reported to stimulate cell migration in different cell types, the molecular mechanisms involved in this process have remained largely elusive. In the present work, we demonstrate that PA stimulates lung adenocarcinoma cell migration and that this action involves an interaction between PA and the LPA1 receptor, and the subsequent activation of mitogen-activated protein kinases (MAPKs). The latter enzymes transmit signals from different stimuli from the cell membrane to the nucleus, thereby stimulating transcriptional factors that promote proliferative or inflammatory responses [13]. Mitogenic agents such as growth factors usually promote ERK1-2 phosphorylation (activation), whereas agents that induce cellular stress, such as proinflammatory cytokines, typically stimulate p38 and/or c-Jun N-terminal kinase (JNK). Another important pathway that is essential in the regulation of cell growth, survival, and migration/invasion processes is the JAK/STAT pathway [14]. The purpose of the present investigation was to determine whether PA could stimulate lung cancer cell migration, and to shed light on the mechanism whereby PA exerts this action. In particular, we focused on the effects of PA on MAPKs and JAK/STAT and showed that ERK1-2, p38, JNK, and the JAK2/STAT3 pathway are implicated in the regulation of lung cancer cell migration by PA.

## 2. Materials and Methods

### 2.1. Materials

The A549 human cell line was purchased from the American Type Culture Collection (ATCC) (Manassas, VA, USA). Dulbecco’s modified eagle’s medium (DMEM) was obtained from Lonza. Fetal bovine serum (FBS) was supplied by Gibco. Phosphatidic acid (from egg yolk lecithin), 1,2-dipalmitoyl-sn-glycero-3-phosphate (16:0 PA), 1-palmitoyl-2-oleoyl-sn-glycero-3-phosphate (16:0-18:1 PA), 1-Oleoyl-sn-glycerol 3-phosphate (Oleoyl-L-α-lysophosphatidic acid), and VPC32183 were from Avanti Polar lipids (Birmingham, AL, USA). Phosphatidic acid, -[glycerol-14C(U)]-, and [oleoyl-9,10-3H]-lysophosphatidic acid were from Perkin Elmer. Fatty-acid free bovine serum albumin (BSA), crystal violet, PD98059, SB202190, SB239063, SP600125, U0126, Ki16425 and PF-8380 (4-[3-(2,3-Dihydro-2-oxo-6-benzoxazolyl)-3-oxopropyl]-1-piperazinecarboxylic acid (3,5-dichlorophenyl)methyl ester were from Sigma-Aldrich. HA-130 (B-[3-[[4-[[3-[(4-Fluorophenyl)methyl]-2,4-dioxo-5-thiazolidinylidene]methyl]phenoxy]methyl]-phenyl]-boronic acid was from Echelon Biosciences Inc. (Salt Lake City, UT, USA). AM966 was from Cayman Chemicals (Ann Harbor, MI, USA). AZD1480, static, the PLA_2_ inhibitors AACOCF3 and PACOCF3, and pertussis toxin were from Tocris Bioscience (Bristol, UK). Tyrphostin A25 was from Santa Cruz (Dallas, TX, USA). Goat anti-rabbit IgG horseradish peroxidase secondary antibody, and antibodies to phospho-ERK1/2 (ca. no. 9101), total ERK1/2 (ca. no. 9102), phospho-p38 (ca.no. 9215), total p38 (ca. no. 9212), phospho-JNK (ca.no. 9251), total JNK (ca.no. 9252), phospho-STAT3 (ca.no. 9131), and total STAT3 (ca.no. 4904) were purchased from Cell Signaling (Danvers, MA, USA).

### 2.2. Cell Culture

The A549 cell line is an alveolar epithelial cell line isolated from a 58-year-old Caucasian male with lung adenocarcinoma. On a regular basis, cells were grown in T75 flasks (surface area 75 cm^2^) in RPMI 1640 culture medium supplemented with 10% heat-inactivated FBS, 50 mg/L gentamicin, and 2 mM L-glutamine. The cells were cultured at 37 °C in a humidified incubator with 5% CO_2_ and were subcultured every 3–4 days, maintaining a cell density between 6000 and 60,000 cells/cm^2^. In some experiments, the cells were preincubated with inhibitors for 90 min before they were challenged with agonists in serum-free RPMI 1640 culture medium supplemented with 0.1% BSA, except for cell migration experiments in which the culture medium was supplemented with 0.2% BSA, as indicated below.

### 2.3. Delivery of PA to Cells in Culture

PA was prepared as an aqueous dispersion in the form of liposomes by sonicating the phospholipid in sterile nanopure water until a clear dispersion was obtained. PA was then added to the cells from 2 mM stock dispersions, as required. No organic solvents were used so as to avoid unwanted side effects.

### 2.4. Cell Viability Assay (Crystal Violet Method)

A549 cells were seeded in 96-well plates at a density of 10^4^ cells/well in a volume of 100 µL in triplicate. Cells were then incubated in RPMI 1640 medium supplemented with 10% FBS overnight. The medium was then replaced with fresh serum-free medium with or without agonists or inhibitors, as required, and further incubated for the required periods of time. The cells were washed twice with PBS and 50 μL of a solution of 0.5% of crystal violet dye in 20% methanol was added to each well. The cells were kept in contact with the dye for 20 min at room temperature while shaking. After staining, the cells were washed three times with distilled water and the plate was left to air-dry for 1–2 h. Then, a volume of 200 μL of methanol was added to each well and the plate was incubated for 20 min at room temperature on a bench rocker, with the lid on. The cells were then washed three times with PBS and left to dry. The crystal violet dye was then eluted with the addition of 200 µL of methanol to each well with shaking for 20 min, and the absorbance was read at 570 nm using a Power Wave XS plate reader from Biotek Instruments (Winooski, VT, USA). 

### 2.5. Determination of Cell Migration

Cell migration was determined using the Boyden chamber cell migration assay (or transwell assay). Twenty-four-well plates containing 12 chemotaxis chambers with filters of 8.0 µm pore diameter (Transwell, Corning Costar) were used in the experiments. The filters were precoated with 30 µL of fibronectin (stock solution 0.2 μg/µL), a glycoprotein used to allow for a better attachment and adhesion of the cells. Cell suspensions (5 × 10^4^ cells in a volume of 100 μL) were then added to the 12 chemotaxis chambers (upper wells) in serum-free RPMI 1640 medium supplemented with 0.2% BSA. Agonists were added to the 12 lower wells diluted in 300 µL of the same medium. When used, inhibitors were added to both the upper and lower wells and cells were preincubated for 90 min prior to the addition of the agonist. Then, the chambers containing the cells were transferred to the lower wells containing the agonist. After the indicated periods of time, the non-migrated cells were removed using a cotton swab, and the migrated cells (that were attached to the opposite side of the filter) were fixed with para-formaldehyde (5% in PBS) for 30 min. Then, the para-formaldehyde was removed and the filters containing the migrated cells were washed twice with PBS. The cells were then stained with a solution of crystal violet (0.1% in PBS) for 20 min. After this time, the dye was removed and the filters were carefully washed with distilled water. The upper sides of the filters were then cleaned again with a cotton swab to remove any remaining non-migrated cells. Filters were then placed on microscope slides and coverslips were sealed with mineral oil, avoiding the formation of bubbles between slides and coverslips. Cell migration was determined by counting the number of migrated cells using a Nikon Eclipse 90i direct light microscope equipped with the NIS-Elements 3.0 software. Alternatively, the migrated cells were counted using a Nikon Eclipse Ts2 inverted light microscope also equipped with NIS-Elements 3.0 software without unmounting the filters. Cell counting was performed in 5 randomly selected microscope fields per filter, at 100× magnification.

### 2.6. Phosphatidic and Lysophosphatidic Acid Analyses

A549 cells (5 × 10^5^ cells/mL) were seeded in 60 mm culture plates in RPMI 1640 culture medium supplemented with 10% FBS and incubated at 37 °C for 24 h. The medium was then replaced with serum-free RPMI 1640 supplemented with 0.1% BSA. After 2 h, the cells were treated with 10 μM [14C]PA (55,500 dpm/plate) for 60 min. The radioactive medium was then removed and kept for lipid extraction. The cells were washed three times with ice-cold Ca^2+^-free PBS (phosphate-buffered saline) and collected in 1 mL of methanol to which 0.5 mL of chloroform was immediately added. Lipid extraction was achieved by the addition of 0.5 mL of chloroform and 0.9 mL of 2 M KCl in 0.2 M H_3_PO_4_, which caused the separation of phases. The chloroform phase was dried down under nitrogen and the lipids were dissolved in 50 µL of a mixture of chloroform–methanol (9:1, *v*/*v*). Lipids were then separated via thin-layer chromatography (TLC) using silica gel 60-coated plates, which were previously soaked in an alcoholic solution of boric acid (2.3% in ethanol) and then developed with a solvent mixture containing chloroform/ethanol/water/triethylamine (30:35:8:35, by vol). After drying, the silica gel plates were stained with sulfuric acid in methanol so as to identify the PA and LPA spots, which were compared with authentic standards that were run alongside the samples. The quantification of the radioactivity in the PA and LPA spots was achieved using liquid scintillation counting.

### 2.7. Western Blotting Analysis

The identification of specific proteins was achieved using Western blotting. Lung cancer A549 cells were seeded in 6-well plates at a density of 2 × 10^5^ cells/well in RPMI 1640 culture medium supplemented with 10% FBS. After 24 h, the medium was replaced with serum-free RPMI 1640 medium and incubated further for 2 h. Agonists or inhibitors were then added as required for the indicated periods of time. Cells were washed twice with cold PBS and harvested in ice-cold lysis buffer containing 0.1 M Tris pH 8, 0.685 M NaCl, 2.5 mM EDTA, 1% (*v*/*v*) igepal, 10% (*v*/*v*) glycerol, and 1μg/mL of Protease Inhibitor Cocktail (PIC). Samples were lysed via sonication prior to the determination of protein concentration, which was determined using the Bradford dye-binding assay from BioRad. Samples ranging from 10 to 40 μg of protein were electrophoresed. To achieve this, the samples were mixed with 4× loading buffer (40% (*v*/*v*) glycerol, 8% sodium dodecyl sulfate, 0.04% (*p*/*v*) bromophenol, 50 μL/mL β-mercaptoethanol, and 240 mM Tris pH 6.8) and heated at 90 °C for 10 min. Samples were then loaded into polyacrilamide gels (15%, 12%, or 7.5% acrylamide) to separate proteins through sodium dodecylsulfate-polyacrilamide gel electrophoresis (SDS-PAGE). Electrophoresis was run in a buffer containing 1.92 glycin, 0.25 M Tris-HCl, and 1% SDS at 125V for 90 min, approximately. Proteins were then transferred onto polyvinylidene difluoride (PVDF) or nitrocellulose membranes. Protein electrotransfer was performed at 400 mA for 75–90 min in ice-cold transfer buffer (14.4 g/L glycin, 3 g/L Tris, and 20% methanol). Before the addition of the primary antibodies, the PVDF or nitrocellulose membranes were blocked for 1 h with 5% skim milk in tris-buffered saline (TBS) containing 0.1% Tween-20, pH 7.6, so as to avoid the unspecific binding of the antibody. After the skim milk was removed, the membranes were washed with TBS and were incubated overnight with the primary antibody diluted in TBS/0.1% Tween-20 with 5% BSA (1:1000) at 4 °C. Then, the membranes were washed three times with TBS/0.1% Tween-20 and incubated with Horseradish Peroxidase (HRP)-conjugated secondary antibody diluted 1:5000–1:10,000 in 1% skim milk in TBS/0.1% Tween-20 for 1 h. Protein bands were visualized via enhanced chemiluminescence using a ChemiDoc™ Imaging System from BioRad and Supersignal West Femto Max as an enhancer.

### 2.8. Statistical Analysis

GraphPad Prism version 8.2.1 software (GraphPad Software, San Diego, CA, USA) was used for statistical analyses. The results are presented as the mean ± standard deviation (SD) of three independent experiments performed in triplicate, unless indicated otherwise. Significant differences were determined using Student’s *t*-test with *p* < 0.05 as statistically significant.

## 3. Results

### 3.1. Phosphatidic Acid Stimulates Human Lung Adenocarcinoma Cell Migration

In the present work, we show that a natural mixture of PA, mainly containing esterified palmitate and oleate as the fatty acid moiety, stimulates the migration of A549 cells, a human lung (alveolar) adenocarcinoma cell line commonly employed to investigate lung cancer growth and dissemination. The upregulation of cell migration was time- and concentration-dependent, with optimal stimulation being attained after 24 h of incubation with 10 µM PA (Figure 1). Similarly, the treatment of the lung cancer cells with different PA species, such as dipalmitoyl PA (C16:0-PA) (Figure 2A) or 1-palmitoyl, 2-oleoyl-PA (C16:0-C18:1-PA) (Figure 2B), stimulated the migration of cancer cells to a similar extent and profile to that of the natural mixture of PA. By contrast, phosphatidylethanol, which is similar to PA but has ethanol as the head group of the phospholipid, failed to stimulate cell migration.

### 3.2. Phosphatidic Acid Induces Phosphorylation of the MAPKs ERK1-2, p38, and JNK. Implication in PA-Stimulated Lung Cancer Cell Migration

To investigate the mechanisms through which PA induces its chemotactic effects in lung cancer, the cells were challenged with an optimal concentration of the phospholipid that causes the maximal stimulation of cell migration, and then the phosphorylation levels of ERK1-2, JNK, and p38, which are major kinases implicated in malignant transformation, were determined [1]. Figure 3A shows that the treatment of lung cancer A549 cells with PA caused a rapid phosphorylation of ERK1-2. To test whether these kinases are involved in the chemotactic effect of PA, the cells were treated with the agonist in the presence of PD98059 (10 µM) or U0126 (3 µM), which are two specific and well-established inhibitors of MEK1-2, the enzyme that phosphorylates ERK1-2. Figure 3B,C shows that both of these inhibitors blocked PA-stimulated cell migration completely at concentrations that are not toxic for the cells (Figure 3D,E), suggesting that MEK1-2/ERK1-2 is an essential pathway in the regulation of lung cancer cell migration.

Like for ERK1-2, the incubation of the cells with PA rapidly increased the phosphorylation of the MAPK p38 (Figure 4A). The implication of this kinase in the chemotactic effect of PA was studied by preincubating the cells with the specific p38 inhibitors SB202190 or SB239063 at concentrations that were not toxic for the cells (10 µM of each inhibitor). Both inhibitors abolished PA-stimulated cell migration (Figure 4B–E), suggesting that p38 is also relevant in this process.

Moreover, PA caused the potent phosphorylation of JNK (Figure 5A) and incubation of the cells with the specific inhibitor SP600125 at concentrations that were not toxic for the cells and blocked PA-stimulated cancer cell migration (Figure 5B,C). Altogether, these data indicate that ERK1-2, p38, and JNK are key regulators of PA-stimulated lung cancer cell migration.

### 3.3. Phosphatidic Acid Stimulates Lung Cancer Cell Migration through Interaction with LPA Receptor 1

Another important finding in this work was that the stimulation of A549 lung cancer cell migration by PA was significantly reduced by the pertussis toxin (PTX) (Figure 6A), a Gi protein inhibitor often used to study the implication of Gi protein-coupled receptors in cell signaling processes. The concentration of PTX used in experiments was 0.2 µg/mL, which was not toxic for the cells (Figure 6B) and was 4-fold higher than the PTX concentration used to completely inhibit the agonist stimulation of Ca^2+^ transients in these same cells [15].

Based on the above observations and our recently published work, in which we demonstrated that PA stimulates cell proliferation through binding to LPA receptors in myoblasts [16], the question was whether PA might also be able to interact with a PTX-sensitive LPA receptor to stimulate lung cancer cell migration. In this connection, real-time quantitative PCR analysis revealed that A549 cells mainly express the PTX-sensitive LPA1 receptor, with only minute amounts of LPA2 and LPA3 being detected [17]. The implication of the LPA1 receptor in the stimulation of lung cancer cell migration by PA was tested by treating the cells with PA in the presence of the well-established LPA1 specific antagonist AM966 [17,18] (Figure 7A), and the also well-established and structurally dissimilar specific LPA1/3 inhibitors Ki1645 [17,19] (Figure 7B), and VPC32183 [19,20] (Figure 7C). The three LPA1 inhibitors were able to completely inhibit PA-stimulated cell migration at concentrations that were not toxic for the cells (Figure 7D,E), suggesting that LPA1 is the receptor mediating the PA stimulation of lung cancer cell migration.

### 3.4. Implication of the JAK2/STAT3 Pathway in the Stimulation of Lung Cancer Cell Migration by Phosphatidic Acid

It is known that cell migration is regulated by protein tyrosine kinases. To evaluate the possible implication of these kinases in the stimulation of cell migration by PA, the selective inhibitor tyrphostin A25 was used. Figure 8 shows that non-toxic concentrations of this inhibitor substantially decreased the stimulation of lung cancer cell migration by PA, suggesting that protein tyrosine kinases were involved in this process. A major pathway that is activated downstream of receptor tyrosine phosphorylation is the JAK2/STAT3 pathway. JAK2/STAT3 signaling is essential for numerous developmental and homeostatic processes, and is involved in the regulation of cell survival, proliferation, inflammation, and the migration/invasion of cancer cells [14].

We show here that PA induces the rapid phosphorylation of STAT3 in the lung cancer cells (Figure 9A). To test whether the JAK2/STAT3 pathway is involved in the chemotactic effect of PA, the cells were treated with the phospholipid in the presence of the specific JAK2 inhibitor AZD1480 (10 µM), or stattic (10 µM), which selectively inhibits STAT3. Figure 9B,C shows that both of these inhibitors blocked the stimulation of cell migration by PA completely, at concentrations that were not toxic for the cells (Figure 9D,E), suggesting that JAK2/STAT3 is a crucial pathway in the regulation of lung cancer cell migration.

### 3.5. The Stimulation of Lung Cancer Cell Migration by PA Is Independent of Conversion to LPA

In a previous study we showed that exogenous PA stimulated C2C12 myoblast proliferation in a manner that was independent of LPA formation, and that the optimal concentrations of PA and LPA to stimulate cell proliferation were similar [16]. LPA is mainly produced at the plasma membrane by the action of PLA2 acting on PA, or by the sequential activation of PLA2 (acting on phosphatidylcholine) and the lysophospholipase D enzyme autotaxin [21], the latter being the most important pathway of LPA formation [22,23,24,25]. In myoblasts, treatment with PA in the presence of the PLA2 inhibitors AACOCF3 (20 µM), PACOCF3 (20 µM), and pyrrolidine (1 µM), at concentrations that fully inhibit PLA2, or with the autotaxin inhibitors H130 or PF8380 (both at 300 nM), did not alter PA-stimulated cell proliferation [16]. Likewise, we observed in this work that none of the other PLA2 inhibitors, at the indicated concentrations, or the autotaxin inhibitors H130 (at 300 nM, a concentration that is over 10-fold higher than its IC_50_ and that completely abolished lymphocyte migration [26]) or PF8380 (300 nM) significantly alter the chemotactic effect of PA (Appendix A). Like for myoblasts, we also found that similar concentrations (10 µM) of PA and LPA are required for the optimal stimulation of lung cancer cell migration (Table 1 and Figure 1, Figure 2, Figure 3 and Figure 4) and that the treatment of these cells with PA did not render significant amounts of LPA. Specifically, the treatment of the lung cancer cells with 10 µM [^14^C]PA (55,500 dpm/10^6^ cells) rendered 220 ± 51 dpm in LPA (mean ± range of two experiments performed in duplicate) after 60 min of incubation. This is about 0.39% of the total number of dpm added to the cells in the form of PA, which is equivalent to 39 nM of LPA. From the data shown in Table 1, it can be inferred that this amount of LPA is not sufficient to significantly stimulate the migration of the lung cancer cells.

## 4. Discussion

Cancer cell growth and dissemination are highly complex processes that are mainly regulated by growth factors and chemokines [27]. However, it is well established that some bioactive lipids, including PA or LPA, are key regulators of signaling pathways associated with tumor progression. In this context, PA levels were shown to be significantly increased in ras- and tyrosine kinase (fps)-transformed fibroblasts, which also had a higher concentration of DAG when compared with control cells [28]. We previously reported that exogenous PA stimulates DNA synthesis and cell proliferation in rat fibroblasts [29], and, more recently, we found that it promoted cell division in mouse myoblasts [16]. Exogenous PA also activated PLD, indicating that external PA can increase the intracellular levels of PA [29]. Moreover, PA has proinflammatory properties, and many of its deleterious effects have been associated with the upregulation of proinflammatory mediators [30]. The latter agents, which include cytokines such as tumor necrosis factor-alpha (TNF-α), interleukin-6, nitric oxide, or prostaglandin E2, have been shown to regulate the migration of different cell types in the context of inflammation and/or cancer dissemination [31,32,33,34,35].

Both PA and LPA can be found in plasma at relatively low micromolar concentrations. Specifically, plasma PA levels have been reported to be around 3.5 µM [36], with similar concentrations of LPA found in the plasma of healthy volunteers [37]. However, it is likely that the local concentrations of these phospholipids may increase under specific circumstances. Interestingly, the intracellular concentration of PA in cardiomyocytes was found to be around 20 µM, and it can be higher in liver cells [38,39]. If these amounts of PA were to be released upon cell activation, local concentrations of extracellular PA (in the range 5–10 µM) would likely be achievable in vivo. Also of interest is the finding that 20 µM PA increases the intracellular concentration of Ca^2+^ through a mechanism involving the direct interaction of PA with LPA receptors [19]. The latter action is compatible with the stimulation of cell proliferation by PA that we observed in rat fibroblasts and mouse myoblasts [16,29]. Also, extracellular vesicles, which can be secreted by cancer cells, are enriched in PA and phospholipase D2, which produces PA [40,41]. These vesicles have been shown to bind to plasma membrane receptors, thereby triggering a variety of cell responses, including tumor invasion and metastasis [42]. Moreover, the release or accumulation of extracellular vesicles at specific extracellular locations might increase the local concentrations of extracellular PA. In addition, PA and LPA can be produced at the plasma membrane of cells by exogenous PLDs (i.e., bacterial PLD) or autotaxin, respectively, thereby increasing their local concentrations as well as their availability for interacting with specific membrane sites, or receptors. In this work, we demonstrate that PA can stimulate lung cancer cell migration and have identified some of the signaling pathways involved in this process. Specifically, the treatment of A549 lung cancer cells with exogenous PA resulted in the stimulation of the MAPKs ERK1-2, p38, and JNK, leading to cell migration. The rapid phosphorylation (activation) of the MAPKs triggered by PA indicates that it may be a receptor-mediated effect. In this connection, mounting evidence suggests that at least some of the biological actions elicited by PA may occur through an interaction with G protein-coupled receptors [43,44,45,46,47]. Noteworthy is that the stimulation of lung cancer cell migration by PA, which we have observed in the present work, could be inhibited by PTX, suggesting the intervention of Gi protein-coupled receptors in the regulation of this process. It has also been reported that PA can stimulate actin polymerization within seconds, an action that leads to the stimulation of human monocyte migration. This effect was optimal after 90 min of PA addition to the cells and was also inhibited by PTX [48]. Also, in the present study, we found that PA elicits its chemotactic effect through an interaction with the LPA1 receptor in the lung cancer cells. In this connection, another study showed that PA induced the haptotactic migration of human monocytes in a receptor-mediated fashion [48]. Although initial studies suggested that the effects of PA might be due to its conversion into LPA, it has been thoroughly demonstrated that PA itself can cause cell activation, and regulate key biological effects including cell proliferation or migration. Specifically, the stimulation of myoblast proliferation by PA involved an interaction between the phospholipid and the LPA1-2 receptors, in the absence of LPA formation [16]. Moreover, contrary to LPA, PA was able to stimulate hair epithelial cell and epidermal keratinocyte growth through mechanisms involving the prior activation of ERK1-2 [49].

Also, in the present work, we found that the incubation of A549 lung cancer cells with PA rendered a very small amount of LPA, which was insufficient to promote cell migration. This finding is consistent with our previous work showing very little conversion of PA into LPA in mouse myoblasts incubated under similar conditions [16]. Moreover, it was reported that PA can stimulate the migration of human neutrophils and leukemia HL-60 monocytes and that these actions could not be recapitulated by LPA [50].

The pathological relevance of the stimulation of lung cancer cell migration by PA through an interaction with the LPA1 receptor resides in the fact that in the absence of LPA formation, the migration of malignant cells may still be possible through the action of PA, a bioactive phospholipid that is formed by enzymes different to those involved in LPA synthesis. In other words, PA may be a substitute for LPA in situations where LPA formation might be compromised in order to stimulate cell migration. This may be an alternative for lung cancer cells to still be able to migrate or invade other tissues in the absence of LPA formation.

Besides stimulating MAPKs, PA has also been shown to control tyrosine phosphorylation of a variety of cellular proteins or transcription factors including STAT3 [51]. In mammalian cells, the JAK/STAT pathway is one of the major mechanisms by which many cytokines and growth factors exert their regulatory actions. It is known that the activation of this pathway leads to an upregulation of various genes that are implicated in cell survival and proliferation [14,52], and we show here that PA stimulates lung cancer cell migration through the JAK2/STAT3 pathway.

Taken together, the results of the present investigation demonstrate that PA stimulates cell migration through a mechanism involving an interaction with the LPA1 receptor and the subsequent activation of the MAPKs ERK1-2, p38, and JNK, as well as the upregulation of the JAK2/STAT3 pathway. These findings point to LPA1/MAPK and JAK2/STAT3 axes as important regulators of human lung cancer cell dissemination, and suggest that targeting PA formation and/or the LPA1 receptor may provide new strategies to reduce malignancy in lung cancer.

## Figures and Tables

**Figure 1 biomedicines-11-01804-f001:**
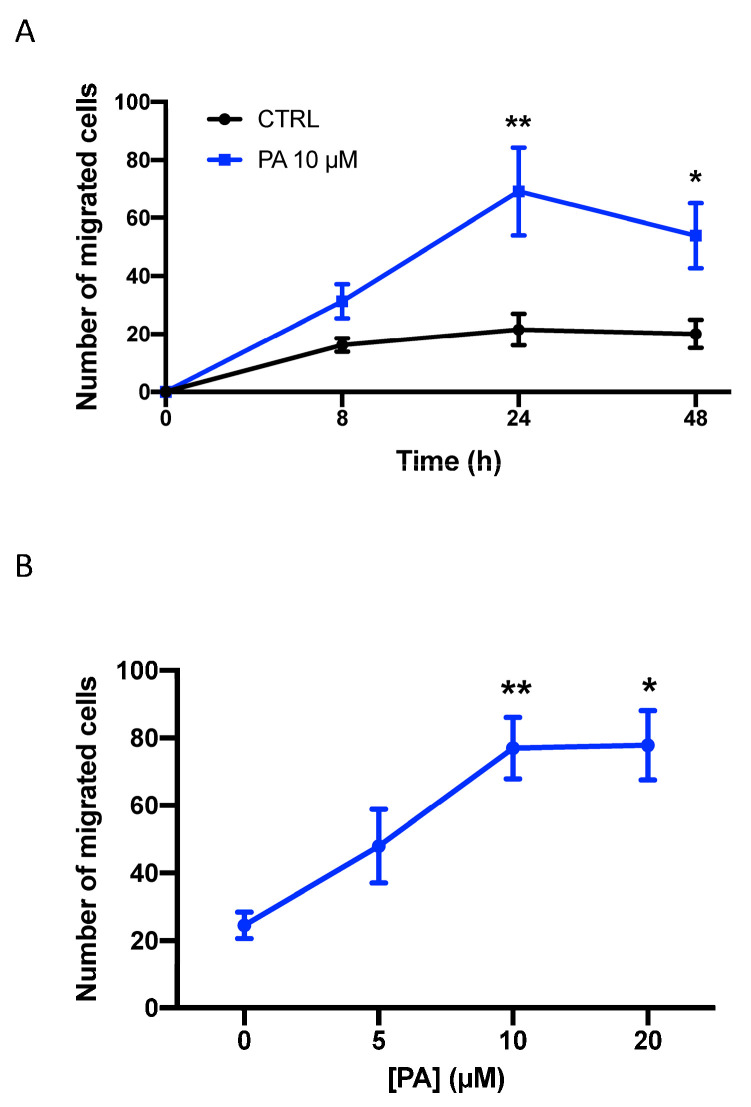
Phosphatidic acid stimulates lung adenocarcinoma cell migration. A549 cells were seeded in Boyden chambers and incubated in serum-free RPMI 1640 culture medium supplemented with 0.2% BSA. Cell migration was determined as indicated in the Materials and Methods section. In (**A**), the cells were incubated in the presence or absence of 10 µM PA for different times, as indicated. In (**B**), the cells were challenged with increasing concentrations of PA for 24 h, as indicated. Data are expressed as mean ± SD of three independent experiments that were carried out in duplicate (* *p* < 0.05; ** *p* < 0.01).

**Figure 2 biomedicines-11-01804-f002:**
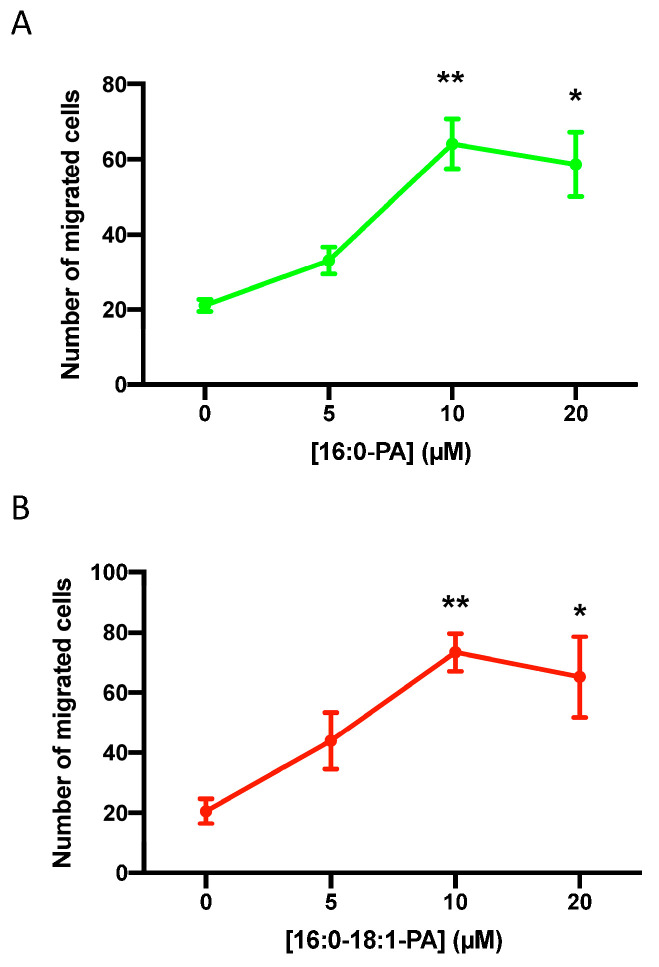
Stimulation of lung adenocarcinoma cell migration via 16:0-PA and 16:0-18:1-PA. A549 cells were seeded and incubated as in Figure 1, and were challenged with 16:0-PA (**A**) or with 16:0-18:1-PA (**B**) at the concentrations that are indicated, for 24 h. Cell migration was determined as indicated in the Materials and Methods section. Data are expressed as mean ± SD of three independent experiments that were carried out in duplicate (* *p* < 0.05; ** *p* < 0.01).

**Figure 3 biomedicines-11-01804-f003:**
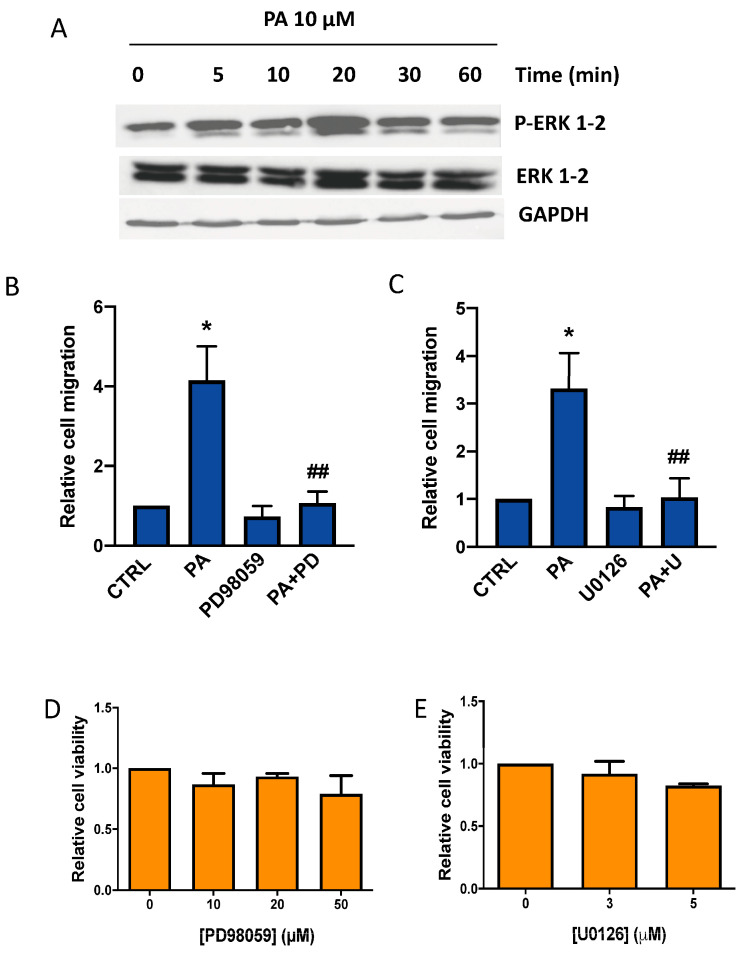
PA induces ERK 1-2 MAPK phosphorylation in lung adenocarcinoma cells. Implication of the MEK/ERK1-2 in the stimulation of lung adenocarcinoma cell migration by PA. (**A**) Incubation of A549 lung adenocarcinoma cells with 10 µM PA caused ERK1-2 phosphorylation. Phosphorylated ERK (P-ERK1-2) were identified via Western blotting analysis using a specific antibody. To monitor for equal loading of protein-specific antibodies to total, ERK1-2 and GAPDH were used, as indicated. Similar results were obtained in an additional experiment. (**B**,**C**) Cells were preincubated for 90 min with or without the specific MEK inhibitors PD98059 (10 µM) or U0126 (3 µM) as indicated, before stimulation with PA (10 µM), and cell migration was measured as detailed in Materials and Methods. Data are expressed relative to the control value and are given as the mean ± SD of 3 independent experiments carried out in duplicate. (* *p* < 0.05, control versus PA-treated cells, ## *p* < 0.01, PA-treated cells versus PA-treated cells in the presence of the inhibitor, PD = PD98059, or U = U0126). (**D**,**E**) Concentrations of PD98059 (10–50 µM) or U0126 (3–5 µM) were not toxic for A549 cells. Cell viability was monitored by staining the cells with crystal violet after 24 h treatment with the inhibitors. Data are expressed relative to the values without the corresponding inhibitor (0 µM) and are given as the mean ± SD of 3 independent experiments carried out in triplicate.

**Figure 4 biomedicines-11-01804-f004:**
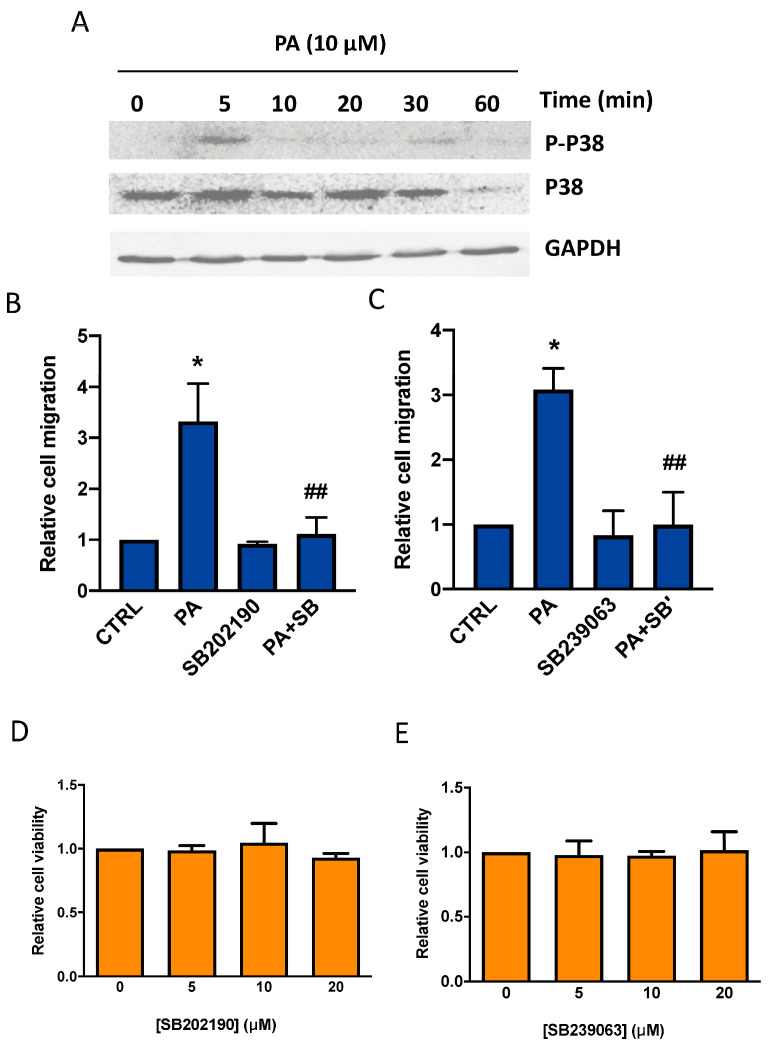
PA induces p38 MAPK phosphorylation in lung adenocarcinoma cells. Implication of p38 MAPK in the stimulation of lung adenocarcinoma cell migration by PA. (**A**) Treatment of A549 lung adenocarcinoma cells with 10 µM PA caused p38 MAPK phosphorylation. Phosphorylated (P) p38 was identified via Western blotting using a specific antibody to P-p38. To monitor for equal loading of protein specific antibodies to total, p38 and GAPDH were used, as indicated. Similar results were obtained in an additional experiment. (**B**,**C**) Cells were preincubated with or without the specific p38 inhibitors SB202190 (10 µM), or SB239063 (3 µM) for 90 min, as indicated, before stimulation with PA (10 µM). Cell migration was measured as indicated in Materials and Methods. Data are expressed relative to the values without the corresponding inhibitor (0 µM) and are given as the mean ± SD of 3 independent experiments carried out in triplicate (* *p* < 0.05, control versus PA-treated cells, ## *p* < 0.01, PA-treated cells versus PA-treated cells in the presence of the inhibitor, SB = SB202190, or SB’ = SB239063). (**D**,**E**) Concentrations of SB202190 (5–20 µM) or SB239063 (5–20 µM) were not toxic for A549 cells. Cell viability was monitored by staining the cells with crystal violet after 24 h treatment with the inhibitors. Data are expressed relative to the values without the corresponding inhibitor (0 µM) and are given as the mean ± SD of 3 independent experiments carried out in triplicate.

**Figure 5 biomedicines-11-01804-f005:**
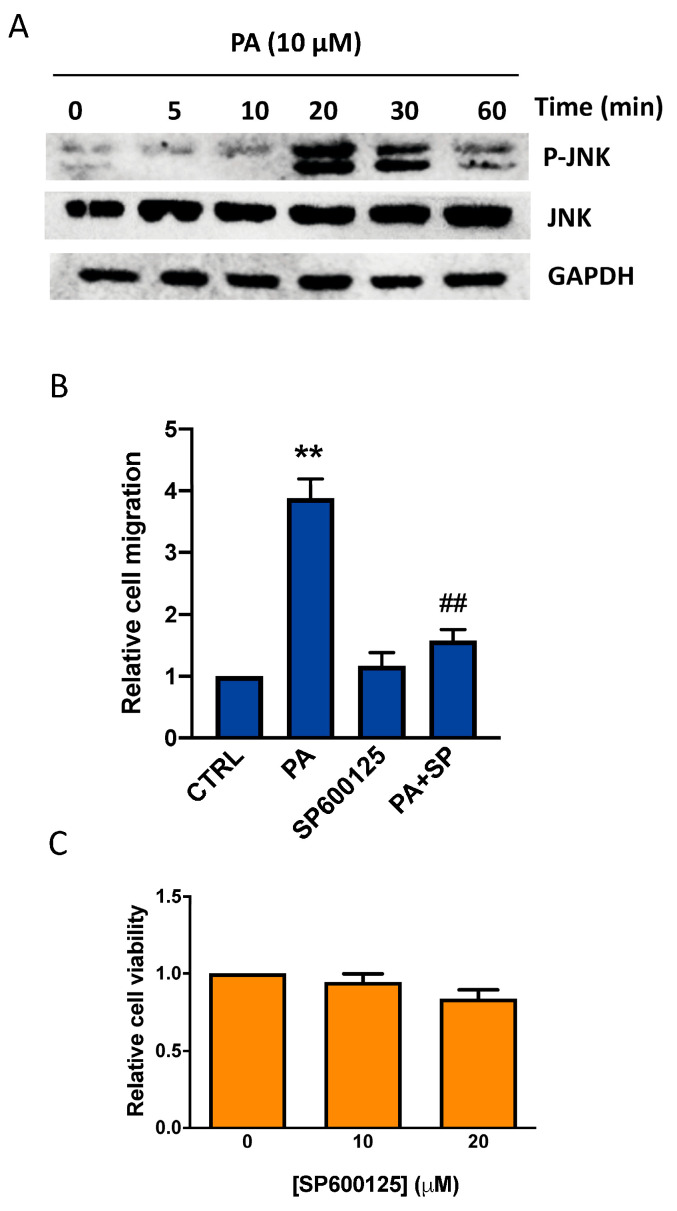
PA induces JNK MAPK phosphorylation in lung adenocarcinoma cells. Implication of JNK MAPK in the stimulation of lung adenocarcinoma cell migration via PA. (**A**) Treatment of A549 lung adenocarcinoma cells with 10 µM PA caused phosphorylation of JNK MAPK. Phosphorylated (P) JNK was identified via Western blotting using a specific antibody to P-JNK. To monitor for equal loading of protein-specific antibodies to total, JNK and GAPDH were used, as indicated. Similar results were obtained in an additional experiment. (**B**) Cells were preincubated for 90 min with or without the specific JNK inhibitor SP600125 (10 µM), before stimulation with PA (10 µM). Cell migration was determined as indicated in Materials and Methods. Data are expressed relative to the control value without agonist and are given as the mean ± SD of 3 independent experiments carried out in duplicate. (** *p* < 0.01, control versus PA-treated cells, ## *p* < 0.01, PA-treated cells versus PA-treated cells in the presence of the inhibitor, SP = SP600125). (**C**) Cell viability was monitored by staining the cells with crystal violet after 24 h treatment with the inhibitor. Concentrations of SP600125 (10–20 µM) were not toxic for A549 cells. Data are expressed relative to the value without inhibitor (0 µM) and are given as the mean ± SD of 3 independent experiments carried out in triplicate.

**Figure 6 biomedicines-11-01804-f006:**
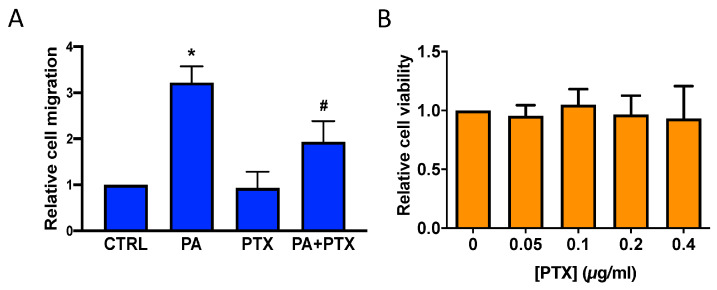
Pertussis toxin inhibits PA-stimulated lung adenocarcinoma cell migration. A549 cells were seeded in Boyden chambers and incubated in serum-free RPMI 1640 culture medium supplemented with 0.2% BSA. (**A**) The cells were preincubated for 90 min with or without PTX (0.2 µg/mL) as indicated, before stimulation with PA (10 µM). Cell migration was determined as indicated in the Materials and Methods section. Data are expressed relative to the control value without agonist and are given as the mean ± SD of 3 independent experiments carried out in duplicate. (* *p* < 0.05, control versus PA-treated cells, # *p* < 0.05, PA-treated cells versus PA-treated cells in the presence of PTX. (**B**) Cell viability was monitored by staining the cells with crystal violet after 24 h treatment with PTX, as indicated in Materials and Methods. PTX (0.05–0.4 µg/mL) was not toxic for A549 cells. Data are expressed relative to the value without inhibitor (0 µM) and are given as the mean ± SD of 3 independent experiments performed in triplicate.

**Figure 7 biomedicines-11-01804-f007:**
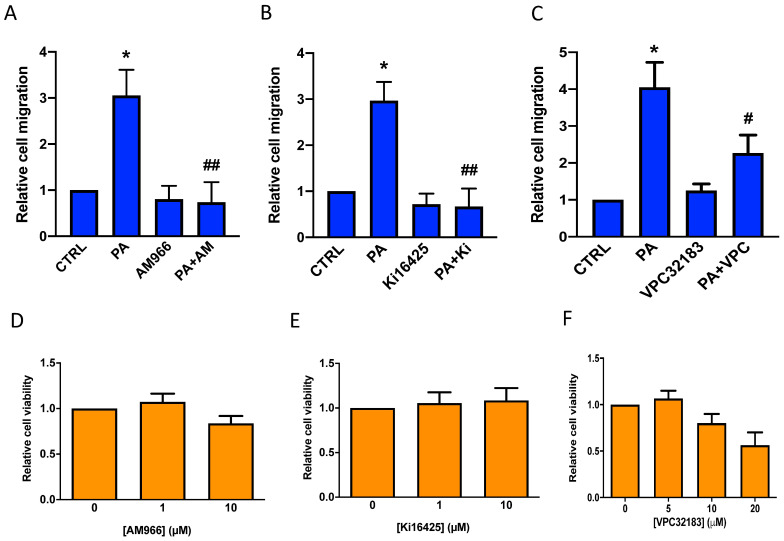
Inhibition of PA-stimulated adenocarcinoma cell migration by the LPA1 receptor antagonists AM966, Ki16425, and VPC32183. A549 cells were seeded in Boyden chambers and incubated in serum-free RPMI 1640 culture medium supplemented with 0.2% BSA. The cells were preincubated for 90 min with or without the LPA1 receptor antagonists AM966 (1 µM) (**A**), Ki16425 (10 µM) (**B**), or VPC32183 (5 µM) (**C**) as indicated, before stimulation with PA (10 µM). Cell migration was measured as indicated in Materials and Methods. Data are expressed relative to the control value without agonist and are given as the mean ± SD of 3 independent experiments carried out in duplicate. (* *p* < 0.05, control versus PA-treated cells; # *p* < 0.05, PA-treated cells versus PA-treated cells in the presence of VPC32183; ## *p* < 0.01, PA-treated cells versus PA-treated cells in the presence of AM966 or Ki16425). AM = AM966, Ki = Ki16425 and VPC = VPC232183. Cell viability was monitored by staining the cells with crystal violet after 24 h treatment with the inhibitors as indicated in Materials and Methods. Concentrations of AM966 (0–10 µM), Ki16425 (0–10 µM), and VPC32183 (5–10 µM) were not toxic for A549 cells (**D**–**F**). Data are expressed relative to the value without the inhibitor (0 µM) and are given as the mean ± SD of 3 independent experiments carried out in triplicate.

**Figure 8 biomedicines-11-01804-f008:**
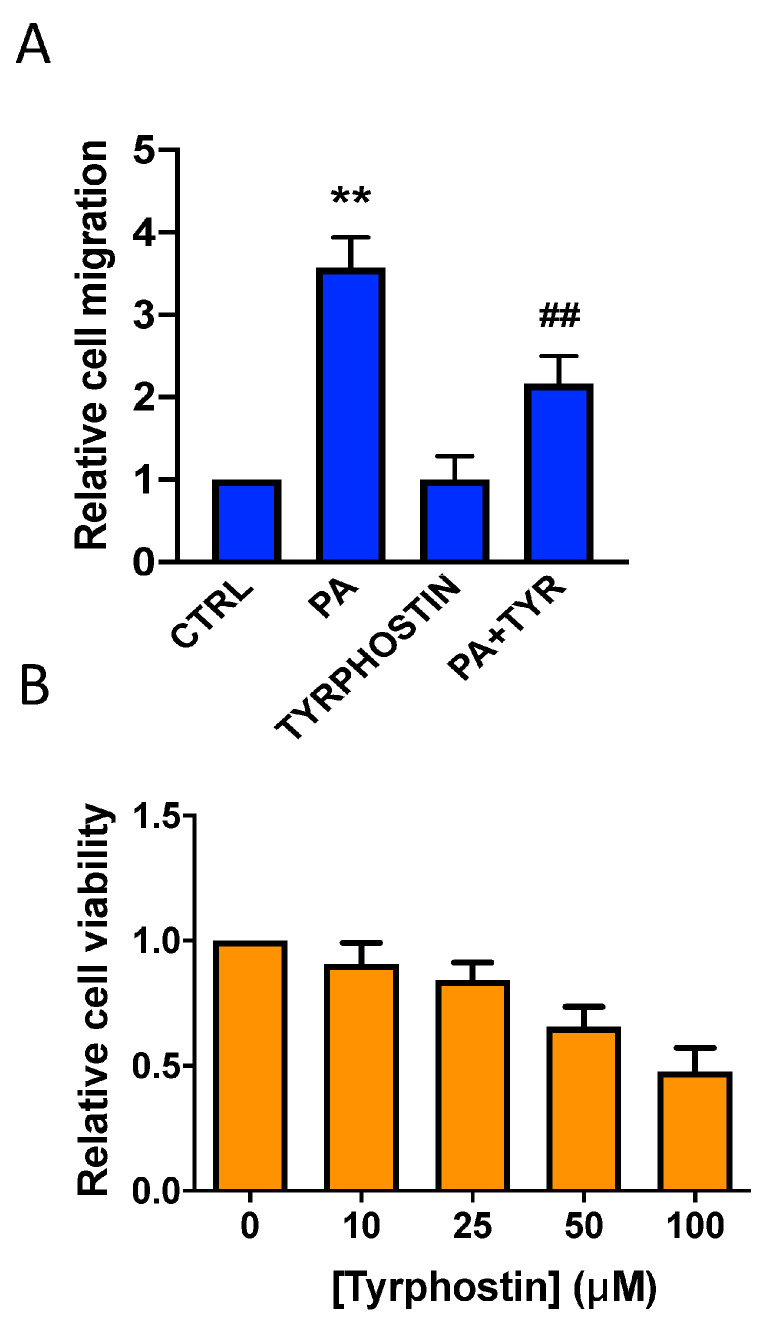
Inhibition of PA-stimulated adenocarcinoma cell migration by the tyrosine phosphorylation inhibitor tyrphostin A25. A549 cells were seeded in Boyden chambers and incubated in serum-free RPMI 1640 culture medium supplemented with 0.2% BSA. (**A**) The cells were preincubated for 90 min with or without the tyrosine phosphorylation inhibitor tyrphostin A25 (25 µM) as indicated, before stimulation with PA (10 µM). Cell migration was measured as indicated in Materials and Methods. Data are expressed relative to the control value without agonist and are given as the mean ± SD of 3 independent experiments carried out in duplicate. (** *p* < 0.01, control versus PA-treated cells; ## *p* < 0.01, PA-treated cells versus PA-treated cells in the presence of tyrphostin A25. TYR = tyrphostin A25). (**B**) Cell viability was monitored by staining the cells with crystal violet after 24 h treatment with the inhibitor, as indicated in Materials and Methods. Concentrations of tyrphostin A25 (10–25 µM) were not toxic for A549 cells. Data are expressed relative to the value without inhibitor (0 µM) and are given as the mean ± SD of 3 independent experiments carried out in triplicate.

**Figure 9 biomedicines-11-01804-f009:**
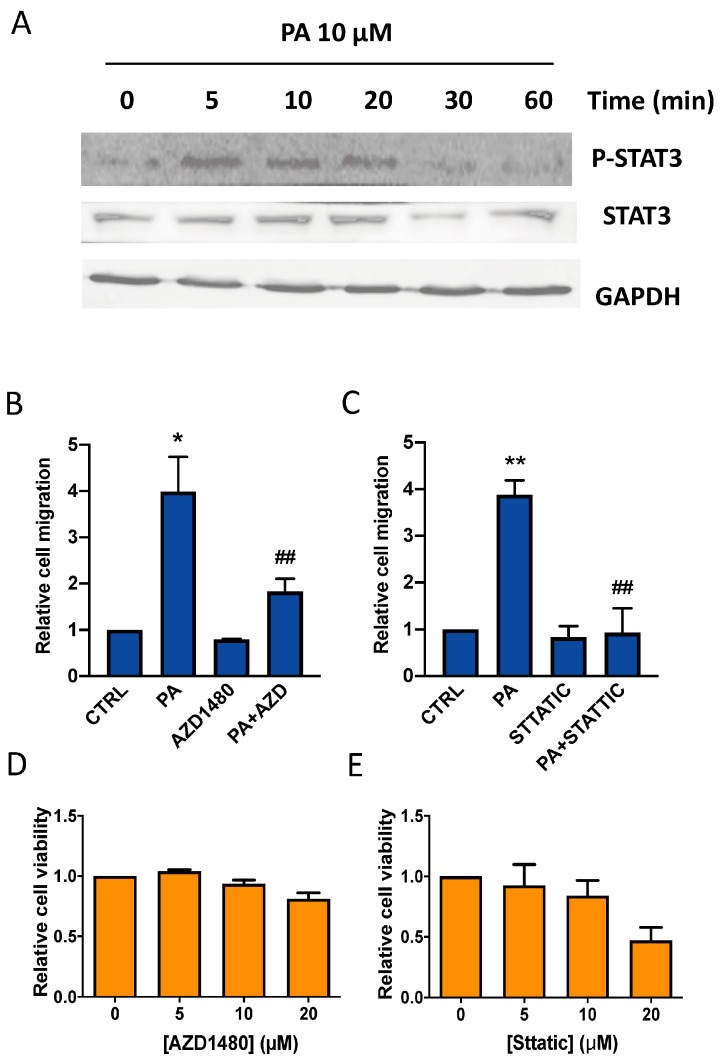
PA induces STAT3 phosphorylation in lung adenocarcinoma cells. Implication of the JAK2/STA3 pathway in the stimulation of lung adenocarcinoma cell migration by PA. A549 cells were seeded in Boyden chambers and incubated in serum-free RPMI 1640 culture medium supplemented with 0.2% BSA. (**A**) Treatment of A549 lung adenocarcinoma cells with 10 µM PA induced rapid phosphorylation of STAT3. Phosphorylated (P) STAT3 was identified via Western blotting using a specific antibody to P-STAT3. To monitor for equal loading of protein-specific antibodies to total, STAT3 and GAPDH were used, as indicated. Similar results were obtained in an additional experiment. (**B**,**C**) Cells were preincubated for 90 min with or without the specific JAK2 inhibitor AZD1480 (10 µM) or the STAT3 inhibitor stattic (10 µM) as indicated, before stimulation with PA (10 µM). Cell migration was determined as indicated in Materials and Methods. Data are expressed relative to the control value without agonist and are given as the mean ± SD of 3 independent experiments carried out in duplicate. (* *p* < 0.05 or ** *p* < 0.01, control versus PA-treated cells, ## *p* < 0.01, PA-treated cells versus PA-treated cells in the presence of inhibitor, AZD = AZD1480). (**D**,**E**) Cell viability was monitored by staining the cells with crystal violet after 24 h treatment with the inhibitors, as indicated in Materials and Methods. Concentrations of AZD1480 (5–20 µM) or stattic (5–10 µM) were not toxic for A549 cells. Data are expressed relative to the value without inhibitor (0 µM) and are given as the mean ± SD of 3 independent experiments carried out in triplicate.

**Table 1 biomedicines-11-01804-t001:** Stimulation of lung cancer cell migration by LPA. A549 lung cancer cells were incubated for 24 h with the concentrations of LPA that are indicated. Cell migration was determined using the Boyden Chamber assay (transwell assay) as detailed in the Materials and Methods section. Data are given as the mean ± SD of three independent experiments carried out in duplicate. *p* values for statistical significance of the chemotactic effect of LPA were calculated by comparison of LPA-treated cells (0.1–10 µM LPA) with non-LPA treated cells (0 µM LPA), as indicated. (N.S. = no significant difference).

[LPA] (µM)	Number of Migrated Cells (Mean ± SD)	Statistical Significance
0	19.2 ± 4.1	
0.1	22.7 ± 7.7	N.S.
1	36.6 ± 9.2	*p* < 0.05
10	72.4 ± 10.0	*p* < 0.01

## Data Availability

All data are available upon request to the corresponding author.

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
