# Peer review of "Phosphatidic Acid Stimulates Lung Cancer Cell Migration through Interaction with the LPA1 Receptor and Subsequent Activation of MAP Kinases and STAT3"

_biomedicines, 2023, doi:10.3390/biomedicines11071804_

Round 1
Reviewer 1 Report
It is known that phosphatidic acid (PA) is a key bioactive glycerophospholipid, which is involved in the regulation of vital cell functions, such as growth, differentiation and migration of cells, as well as in many pathological processes. However, the molecular mechanisms by which PA exerts its pathophysiological effects are not fully understood. In the present work, we have demonstrated that PA stimulates the migration of A549 human non-small cell lung cancer (NSCLC) adenocarcinoma cells, as determined by a well migration assay. The authors argued that PA stimulates lung adenocarcinoma cell migration through interaction with the LPA1 receptor and subsequent activation of MAPKs ERK1-2, p38, and JNK, and that the JAK2/STAT3 pathway is also important in this process. These data suggest that influencing the formation of PA and/or the LPA1 receptor may provide new strategies to reduce malignancy in lung cancer.
I really liked the article, the material is logically and consistently presented, well structured, the design of the study raises no questions. I think that the article certainly deserves publication.
Author Response
ANSWER: We thank the reviewer for the positive comments and compliments about our work, and for acknowledging that the article deserves publication.

Reviewer 2 Report
The authors studied the pathophysiological role of phosphatidic acid (PA) in human non-small cell lung cancer (NSCLC) A549 adenocarcinoma cells. Specifically, stimulation of cell migration by PA using a transwell assay. Stimulation fo cell migration by PA has been examined in earlier studies, though the molecular mechanism is not fully understood.
The authors show that PA stimulate A549 cell migration (using Boyden chamber assay). They then show that PA treatment leads to rapid phosphorylation of ERK1/2 and MEK1-2 inhibitors (PD98059, U0126) blocked PA-stimulated cell migration, indicating that PA-dependent ERK1/2 phosphorylation is directly involved in the cell migration stimulation. Further experiments suggest also involvement oy JAK/STAT3 pathway.
Comments
Results
Section 3.5 (line 415-418):
Because lyso-PA can also stimulate cell migration (as also shown by the authors in Table 1), it was important to check whether formation of lyso-PA from PA was responsible for stimulation of cell migration: according to the authors PLA2 inhibitors did not affect the PA stimulation of cell proliferation of myoblasts:
It is not clear whether this refers to previous reports by the authors (if so give references) or experiments performed within the present study (if so, show results, at least in supplementary information).
line 418-422:
The authors wrote "that none of the later PLA2 inhibitors [...] did not significant alter the chemotactic effect of PA (data not shown)".
(1) Is this a writing error? do they mean that none of the inhibitorsdid significantly alter the chemotactic affect of PA"?
(2) The data should be shown.
Figures 3A, 4A, 5A, 9A show only one western blot, there is no quantification; how many independent experiments have been performed? Usually, three independent experiments should be done; and if so, the authors could also show a quantification of all experiments and perform a statistical analysis.
Phospho-ERK1/2 and total ERK1/2 shown in the "original images" file appear to be not from the same blot; However, it would be much better to successively probe the same blot with the two antibodies to be able to exactly calculate the relative ERK phosphorylation;
Moreover, panel 3A P-ERK-1/2 in the "original images" does not show the complete blot (in contrast to total ERK-1/2 and GAPDH)-why?
Much more importantly: the total ERK-1/2 blot shown in "original images" ist not identical to the blot shown in Figure 3A of the manuscript!
Figure 9A: The whole P-STAT3 and total STAT3 blots should be shown in the "original images"-file
Author Response
REVIEWER 2:
The authors studied the pathophysiological role of phosphatidic acid (PA) in human non-small cell lung cancer (NSCLC) A549 adenocarcinoma cells. Specifically, stimulation of cell migration by PA using a transwell assay. Stimulation of cell migration by PA has been examined in earlier studies, though the molecular mechanism is not fully understood.
The authors show that PA stimulate A549 cell migration (using Boyden chamber assay). They then show that PA treatment leads to rapid phosphorylation of ERK1/2 and MEK1-2 inhibitors (PD98059, U0126) blocked PA-stimulated cell migration, indicating that PA-dependent ERK1/2 phosphorylation is directly involved in the cell migration stimulation. Further experiments suggest also involvement of JAK/STAT3 pathway.
Comments
Results
Section 3.5 (line 415-418):
Because lyso-PA can also stimulate cell migration (as also shown by the authors in Table 1), it was important to check whether formation of lyso-PA from PA was responsible for stimulation of cell migration: according to the authors PLA2 inhibitors did not affect the PA stimulation of cell proliferation of myoblasts:
It is not clear whether this refers to previous reports by the authors (if so give references) or experiments performed within the present study (if so, show results, at least in supplementary information).
ANSWER: We demonstrate in this work that exogenous PA does not result in significant conversion to LPA in A549 lung adenocarcinoma cells. Specifically, we treated the lung cancer cells with radioactive PA (10 µM) and only found a 0.39 % conversion of PA into LPA after 60 min of incubation. This percentage of LPA is equivalent to 39 nM, which according to the results presented in Table 1, is not sufficient to stimulate cell migration. Also, it should be borne in mind that stimulation of ERK1-2, p38, JNK and STAT3 phosphorylation by PA takes place very rapidly (5-20 min), and in any case phosphorylation of these proteins by PA occurs at much earlier time points than 60 min, with no LPA formation. The latter results indicate that PA causes phosphorylation of MAPKs and STAT3 independently of conversion into LPA. These results were already included in Section 3.5 (new lines 435-441).
The sentence stating that “PLA2 inhibitors did not affect the PA stimulation of cell proliferation of myoblasts” refers to our previously published data. The appropriate reference (# 16) is now included after this sentence.
line 418-422:
The authors wrote "that none of the later PLA2 inhibitors [...] did not significant alter the chemotactic effect of PA (data not shown)".
(1) Is this a writing error? do they mean that none of the inhibitors did significantly alter the chemotactic affect of PA"?
(2) The data should be shown.
ANSWER: (1) We thank the reviewer for detecting this error. We apologize for this oversight. What we meant is that none of the PLA2 or autotaxin inhibitors was able to significantly alter the chemotactic effect of PA. This has now been corrected in the text. (2) Following the reviewer´s request we have included the data as supplementary figure 1.
Figures 3A, 4A, 5A, 9A show only one western blot, there is no quantification; how many independent experiments have been performed? Usually, three independent experiments should be done; and if so, the authors could also show a quantification of all experiments and perform a statistical analysis.
ANSWER: Figures 3A, 4A, 5A and 9A show one representative western blot out of two, so as to confirm that PA can stimulate phosphorylation of these proteins. These observations have been previously reported by different groups, including our own, in different cells types (i.e, ref 16). However, the novelty of our study was the demonstration that ERK1-2, p38, JNK, and STAT3 are implicated in the chemotactic effect of PA, for which we performed three independent experiments using various specific inhibitors of each pathway (Figures 3, 4, 5, and 9), and applied appropriate statistical analyses.
Phospho-ERK1/2 and total ERK1/2 shown in the "original images" file appear to be not from the same blot; However, it would be much better to successively probe the same blot with the two antibodies to be able to exactly calculate the relative ERK phosphorylation;
ANSWER: We agree with the reviewer. However, in our hands, reprobing western blots with different antibodies after western blot membrane stripping does not always work appropriately. In this specific situation, we run a second gel using exactly the same sample, under exactly the same conditions. Total ERK bands were consistently similar, showing similar intensities throughout the whole membrane, also indicating that protein loading was equal for each condition.
Moreover, panel 3A P-ERK-1/2 in the "original images" does not show the complete blot (in contrast to total ERK-1/2 and GAPDH)-why?
ANSWER: Most of the times we cut the western blot membranes into long pieces (strips) so as to use different antibodies for the different proteins under study. In this particular case for total ERK1-2 and GAPDH we did not use antibodies other than those for ERK and GAPDH, so there was no need to cut the membranes into strips.
Much more importantly: the total ERK-1/2 blot shown in "original images" ist not identical to the blot shown in Figure 3A of the manuscript!
ANSWER: The reason for this has been explained above.
Figure 9A: The whole P-STAT3 and total STAT3 blots should be shown in the "original images"-file
ANSWER: The western blot membranes for identification of P-STAT3 and total STAT3 were cut into strips for probing with specific antibodies against these proteins, leaving the rest of the membrane available for identification of other proteins. This is the reason why we show membrane strips, not the whole western blot membrane.

Reviewer 3 Report
Evaluation of the manuscript entitled “Phosphatidic acid stimulates lung cancer cell migration through interaction with the LPA1 receptor and subsequent activation of MAP kinases” written by Ana Gomez-Larrauri and co-authors sent to Biomedicines.
1. What is the main question addressed by the research?
The authors tried to take a deeper look at molecular mechanism related to phosphatidic acid (PA) pathophysiological actions. In details, the authors focused on the role of PA in the stimulated migration of the human non-small cell lung cancer (NSCLC) A549 adenocarcinoma cells. This is a short cut of a wide range of possible PA actions but in my opinion our knowledge gets forwards via small steps. I admire the authors’ approach. Good work.
2. Do you consider the topic original or relevant in the field? Does it address a specific gap in the field?
Although the topic is not entirely original as the literature provides some knowledge about PA and its role in some diseases, there is still a gap about specific mechanisms how PA really acts.
3. What does it add to the subject area compared with other published material?
The authors clearly showed that PA stimulates lung adenocarcinoma cell migration through interaction with the LPA1 receptor and subsequent activation of the MAPKs ERK1-2, p38 and JNK, and that the JAK2/STAT3 pathway is of paramount importance in this process.
4. What specific improvements should the authors consider regarding the text?
At the end of the Introduction section the authors must provide a clear hypothesis and/or what they were supposed to discover by the experiments conducted.
Statistics and its presentation: the authors stated that they used one-way ANOVA or t-test as appropriate. Figure 1 and figure 2: what comparison was made? You showed (*p < 0.05; **p < 0.01) in the figures but what was compared to? Figure 3: (*p<0.05, control versus PA-treated cells, ##p <0.01, PA-treated cells versus PA-treated cells in the presence of the inhibitor) – OK but here you compare two groups, so the t-test was used. OK. Let me know when and where one-way ANOVA was applied? Figures 4, 5, 6, 7, 8, 9 – same query as to figure 3.
5. Are the conclusions consistent with the evidence and arguments presented and do they address the main question posed?
The conclusions are right and justified by the main text.
6. Are the references appropriate?
The references were used in an appropriate manner.
Author Response
REPONSES TO REVIEWER 3:
Comments and Suggestions for Authors
Evaluation of the manuscript entitled “Phosphatidic acid stimulates lung cancer cell migration through interaction with the LPA1 receptor and subsequent activation of MAP kinases” written by Ana Gomez-Larrauri and co-authors sent to Biomedicines.
- What is the main question addressed by the research?
The authors tried to take a deeper look at molecular mechanism related to phosphatidic acid (PA) pathophysiological actions. In details, the authors focused on the role of PA in the stimulated migration of the human non-small cell lung cancer (NSCLC) A549 adenocarcinoma cells. This is a short cut of a wide range of possible PA actions but in my opinion our knowledge gets forwards via small steps. I admire the authors’ approach. Good work.
ANSWER: We thank the reviewer for this praise and positive comment about our work
- Do you consider the topic original or relevant in the field? Does it address a specific gap in the field?
Although the topic is not entirely original as the literature provides some knowledge about PA and its role in some diseases, there is still a gap about specific mechanisms how PA really acts.
ANSWER: We agree with the reviewer on that the mechanisms whereby PA exerts its effects are still incomplete. There are just a few papers in the literature showing that PA stimulates cell migration. However, no study has ever shown that PA stimulates lung cancer cell migration and no molecular mechanism for this action has ever been described. Our work demonstrates for the first time that PA stimulates lung cancer cell migration, and have established that the molecular mechanism for this action involves interaction of PA with LPA receptor 1 (LPA1) and subsequent activation of the MAP kinases ERK1-2, p38 and JNK, and the JAK2/STAT3 signaling pathway.
- What does it add to the subject area compared with other published material?
The authors clearly showed that PA stimulates lung adenocarcinoma cell migration through interaction with the LPA1 receptor and subsequent activation of the MAPKs ERK1-2, p38 and JNK, and that the JAK2/STAT3 pathway is of paramount importance in this process.
ANSWER: We thank the reviewer for this positive comment about our work
- What specific improvements should the authors consider regarding the text?
At the end of the Introduction section the authors must provide a clear hypothesis and/or what they were supposed to discover by the experiments conducted.
ANSWER: Following the reviewer´s indications, we have included a new sentence at the end of the Introduction section to address this point.
Statistics and its presentation: the authors stated that they used one-way ANOVA or t-test as appropriate. Figure 1 and figure 2: what comparison was made? You showed (*p < 0.05; **p < 0.01) in the figures but what was compared to? Figure 3: (*p<0.05, control versus PA-treated cells, ##p <0.01, PA-treated cells versus PA-treated cells in the presence of the inhibitor) – OK but here you compare two groups, so the t-test was used. OK. Let me know when and where one-way ANOVA was applied? Figures 4, 5, 6, 7, 8, 9 – same query as to figure 3.
ANSWER: The reviewer is absolutely right. In the present study we have always compared two groups of data for statistical calculations (control versus agonist-treated cells, or PA-treated cells versus PA-treated cells in the presence of the corresponding inhibitor), and have only used t-test to determine if the differences were statistically significant. We have corrected this in section 2.8.
- Are the conclusions consistent with the evidence and arguments presented and do they address the main question posed?
The conclusions are right and justified by the main text.
ANSWER: We thank the reviewer for acknowledging that the conclusions are right and justified by the main text.
- Are the references appropriate?
The references were used in an appropriate manner.
ANSWER: We thank the reviewer for acknowledging that the references were used in an appropriate manner.

Round 2
Reviewer 2 Report
The following point was addressed satisfactory:
"Much more importantly: the total ERK-1/2 blot shown in "original images" ist not identical to the blot shown in Figure 3A of the manuscript!
ANSWER: The reason for this has been explained above."
The original images should display the original blots used for preparing the Figures in the manuscript: I cannot see, why this should not be possible here. In their answer the authors did not explain this.
Author Response
REVIEWER:
"Much more importantly: the total ERK-1/2 blot shown in "original images" is not identical to the blot shown in Figure 3A of the manuscript!
The original images should display the original blots used for preparing the Figures in the manuscript: I cannot see, why this should not be possible here. In their answer the authors did not explain this.
ANSWER: The reviewer is absolutely right. I apologize for having misunderstood this point. We mistakenly included the original image of total ERK1-2 corresponding to the replicate experiment, not to the one corresponding to the blot shown on Figure 3A.
We are now attaching the original images of the blots shown on Figure 3A.
We very much appreciate the insistence of the reviewer on this issue, which helped us to clarify the point.
